# How do social networks, perception of social isolation, and loneliness affect depressive symptoms among Japanese adults?

Natsuho Kushibiki[1], Miyuki Aiba[2], Haruhiko Midorikawa[3], Kentaro Komura[4], Daichi Sugawara[5], Yuki Shiratori[6], Naoaki Kawakami[5], Takafumi Ogawa[7], Chie Yaguchi[1], Hirokazu Tachikawa[1]*

1 Department of Disaster and Community Psychiatry, Institute of Medicine, University of Tsukuba, Tsukuba, Ibaraki, Japan, 2 Faculty of Human Sciences, Toyo Gakuen University, Bunkyo, Tokyo, Japan, 3 Department of Psychiatry, University of Tsukuba Hospital, Tsukuba, Ibaraki, Japan, 4 Faculty of Humanities and Social Sciences, Hirosaki University, Bunkyo, Hirosaki, Aomori, Japan, 5 Faculty of Human Sciences, University of Tsukuba, Tsukuba, Ibaraki, Japan, 6 Department of Psychiatry, Institute of Medicine, University of Tsukuba, Tsukuba, Ibaraki, Japan, 7 Ibaraki Prefectural Medical Center of Psychiatry, Asahimachi, Kasama, Ibaraki, Japan

* tachikawa@md.tsukuba.ac.jp

**Data Availability Statement:** All relevant data are within the manuscript and its Supporting Information files.

## Abstract

### Objective

This study aims to elucidate the complex relationship among social isolation, loneliness, and perception of social isolation and its influence on depressive symptoms by evaluating a hypothetical model. This understanding is essential for the formulation of effective intervention strategies.

### Methods

We conducted an online survey on Japanese adults ($N$ = 3,315) and used the six-item Lubben Social Network Scale to assess the size of their social networks. We employed a single question to gauge their perception of social isolation. Loneliness was assessed using the three-item UCLA Loneliness Scale, and depressive symptoms were examined using the Patient Health Questionnaire-9. Structural equation modeling was employed to test the hypothesized model.

### Results

The final model demonstrated satisfactory fit with data ($\chi^2$ (1) = 3.73; not significant; RMSEA = 0.03; CFI = 1.00; TLI = 1.00). The size of social network demonstrated a weak negative path to loneliness and depressive symptoms ($\beta$ = −.13 to −.04). Notably, a strong positive association existed between perception of social isolation and loneliness ($\beta$ = .66) and depressive symptoms ($\beta$ = .27). Additionally, a significant positive relationship was found between loneliness and depressive symptoms ($\beta$ = .40). Mediation analysis indicated that perception of social isolation and loneliness significantly intensified the relationships between social networks and depressive symptoms.

**Funding:** This work was supported by JST RISTEX "SOLVE for SDGs: Preventing Social Isolation & Loneliness and Creating Diversified Social Networks" Grant Number JPMJRX21K2, Japan. The funders had no role in study design, data collection and analysis, decision to publish, or preparation of the manuscript.

**Competing interests:** The authors have declared that no competing interests exist.

## Conclusions

Results indicate that interventions of psychological approaches, such as cognitive–behavioral therapy, are effective in reducing the perception of social isolation and loneliness, which may lead to the prevention of depressive symptoms. Future longitudinal studies are expected to refine and strengthen the proposed model.

## Introduction

Social isolation and loneliness have emerged as prominent global societal challenges. Initiatives, such as the designation of a *Minister of Loneliness* in 2018 in the United Kingdom, which underscore the severity of the issue, serves as evidence of this fact. In Japan, the phenomenon of *hikikomori*, in which individuals exhibit extreme social withdrawal, epitomizes this challenge. A particularly acute issue, which is known as the *8050 problem*, involves octogenarian parents caring for their socially reclusive children aged in their 50s [1]. In this problem, socially withdrawn children become middle-aged, and the parents who have cared for them typically become elderly, such that caring for their children becomes increasingly difficult. This issue is not unique to Japan; instead, it is a grave one internationally [2] and measures are required to address this issue. Factors, such as social anxiety, avoidant personality disorder, and modern depression, are increasingly linked to increased social isolation and loneliness in Japan [3–5], where these issues are more pronounced compared with those in other cultures [6, 7]. Specifically, social anxiety and avoidant personality disorders are considered to lead to social withdrawal, especially among young people [8]. Additionally, the increase rise of individuals *Not in Employment, Education or Training* (NEET) and unemployment-related economic challenges are associated with increased social isolation and loneliness [9, 10]. The COVID-19 pandemic has exacerbated this situation, which emphasizes the need for assessment and intervention strategies that are globally effective [11].

### Social isolation and loneliness

Although social isolation and loneliness are related, they are distinct concepts that need to be clearly distinguished [12]. Social isolation refers to the *objective* state of having limited social contact with others, including family, friends, and the broad community [13]. This condition is marked by factors such as the number of social ties of a person (e.g., marital status, circle of friends, and family connections) [14, 15], living alone [16], and frequency of interactions with family members [17]. Conversely, a widely used definition of loneliness is the lack of social connections (social loneliness) or the presence of negative feelings (emotional loneliness) that emerge when the quantity or quality of relationships with particular partners/peers is *subjectively* deficient compared with one's ideal [13]. Both conditions adversely affect health but in distinct ways [18]. Social isolation can negatively impact health due to the absence of support and neglect of healthy behaviors [19]. At the same time, loneliness can lead to health decline through psychosomatic pathways, which manifests in decreased self-esteem, a diminished sense of social support, and an increased negative mood [20, 21]. Interestingly, research has demonstrated only weak to moderate correlations between social isolation and loneliness [22], suggesting that having numerous social connections does not automatically alleviate feelings of loneliness [23]. Hence, addressing social isolation and loneliness as separate entities is essential for an accurate assessment and understanding [14].

## Perception of social isolation

The objective state of social isolation and how a person subjectively perceives their isolation are essentially different. Research indicates that one's perceptions of the size of one's social circle are closely linked to experiences of loneliness and social isolation [24]. Notably, perceived social isolation can lead to loneliness, which significantly impacts health than objective social isolation can [25]. Therefore, when assessing health outcomes related to social isolation and loneliness, evaluating the objective (i.e., size of social networks) and subjective perceptions of social isolation is crucial.

## Social isolation, loneliness, and depressive symptoms

Depressive symptoms rank among the most significant health consequences of social isolation and loneliness. Social isolation, which acts as a direct stressor, leads to increased stress responses in the brain [20], and is identified as a partial cause of long-term depression [26]. Research demonstrates that factors that lead to depression due to social isolation include living alone, having a weak social network, and limited social interaction [27, 28]. Loneliness has a well-established connection with depression [29] and is even considered a stronger predictor of depression than the objective measure of social connectedness [30, 31]. These studies indicate that deep-stated loneliness is associated with severe depressive symptoms across age groups [30, 31]. Additionally, perceived social isolation and the lack of societal support can exacerbate depressive symptoms, which complicates the recovery of depressed individuals [32]. Although numerous studies explored the link among social isolation, loneliness, and perception of social isolation, only a few examine these factors as distinct concepts that sequentially influence depression. Scholars propose different treatment and intervention approaches for social isolation and loneliness [12, 33, 34]. However, effectively tailoring treatments and interventions to the specific characteristics of social isolation, loneliness, and perception of social isolation requires a thorough investigation of the contribution of factors to depressive symptoms and their influence on one another.

## The present study

This study intends to elucidate the relationships and underlying mechanisms that link social isolation, perception of social isolation, loneliness, and depressive symptoms. Drawing on prior research, the current study developed a hypothetical model (Fig 1) for exploring the influence of social isolation (assessed by the size of social networks), perception of social isolation, and loneliness on depressive symptoms. On the basis of the existing literature, we hypothesize that increasing one's social networks is negatively related to perception of social isolation, loneliness, and depressive symptoms [24, 27, 28]. We expect that the perception of social isolation, which emerges from the evaluation of individuals of their social networks, is positively related to loneliness and depressive symptoms [24, 25, 32]. Consistent with its established link to depression [29], we propose that loneliness is positively related to depressive symptoms. Furthermore, we argue that the perception of social isolation and loneliness potentially mediates the relationship between social networks and depressive symptoms, because this perception frequently exerts a significant impact on mental health than do measures of objective social isolation [25, 30, 31]. Although earlier studies on social isolation and loneliness primarily focus on older adults [35], we recognize that these issues can affect individuals at any stage of life [30, 31] and broaden the scope by including other age groups [36, 37].

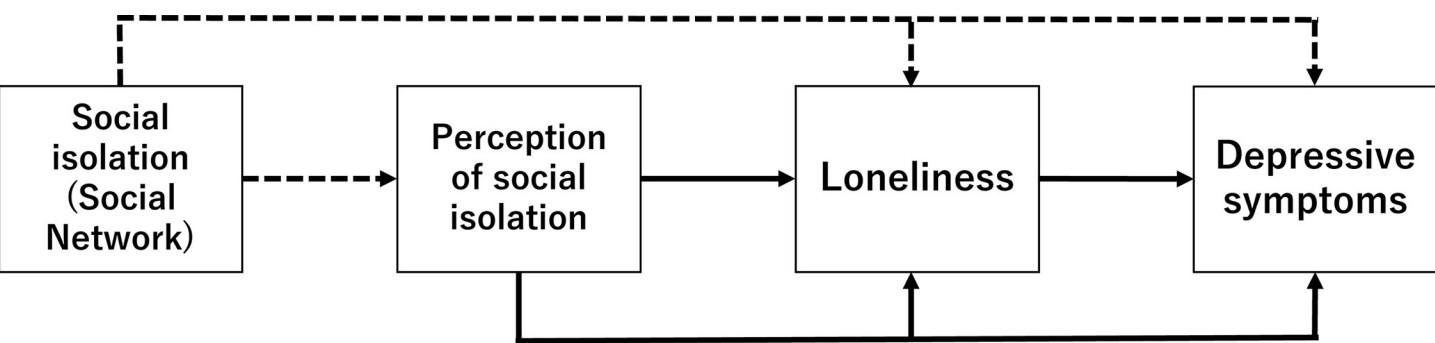

**Fig 1. Hypothetical model.** In this diagram, squares represent different psychological variables. Arrows between these squares depict the relationships among these variables. Dashed arrows illustrate negative paths, while solid arrows indicate positive ones.

## Method

### 1. Participants and procedures

This research was conducted from March 23 to 28, 2022, in partnership with Cross Marketing Inc. We opted to collaborate with Cross Marketing Inc, because it is the most prominent research company in Japan. Sample size was determined using G*Power 3.1.9.7 [38, 39]. The objective of this study was to explore a hypothetical model, which requires a sample size of 779 based on a two-tailed test with an assumed correlation coefficient of 0.10, a significance level of 5%, and a power of 80%. We also conducted structural equation modeling (SEM). However, no consensus was reached on precalculating the power and sample size for SEM [40]. Scholars recognized that a small sample size may compromise the accuracy and reproducibility of SEM results [41]. Therefore, we aimed to use the largest feasible sample size within our funding constraints, which exceeds 779 participants, to ensure a robust analysis and to capture trends representative of the Japanese population. The participants were recruited through a website through the 3.55 million monitor base of Cross Marketing Inc., which targeted Japanese individuals aged 20 years and above. This recruitment procedure ensured targeting a broad spectrum of respondents throughout Japan. The survey was considered complete after receiving 3,500 responses.

We excluded entries from the dataset with missing or incomplete information to ensure the quality and integrity of the data. Prior to the survey, the participants were informed that their responses will remain anonymous and that no negative consequences will be incurred for opting to withdraw at any point or not to respond. The researchers explained that all responses would be processed statistically, which ensures no individual identification. Furthermore, the study followed strict protocols in handling all responses, from the processing stage to data storage and eventual disposal. Participation was entirely voluntary and based on informed consent. They received reward points from the survey agency as a gesture of gratitude for their time and input. These points, which were determined by the number of questions answered, are a unique feature of Cross Marketing and could be exchanged by participants for goods or cash.

The criteria for exclusion included responses that were evidently dishonest (e.g., identical numerical answers for all items) and missing responses to the scales for measuring social networks, perception of social isolation, loneliness, and depressive symptoms. Therefore, we analyzed 3,315 valid responses (men: 1,805, women: 1,493, other genders: 7, and undisclosed gender: 10; average age: 50.05 years, $SD$ = 10.69). The research ethics committee of Toyo Gakuen University approved the study (approval number 2021–010).

## 2. Measures

**Social networks of participants.**   We utilized the Lubben Social Network Scale (LSNS-6) [42] to evaluate the extent of social networks. This scale assesses social isolation from friends, neighbors, and family by examining three aspects, network size, closeness of contacts, and perceived availability of help [43]. The LSNS-6 consists of six items, which are equally divided into three items each for family and friends/neighbors [44]. Items were rated using a six-point scale ranging from 0 (*none*) to 5 (*9 or more*), in which high scores indicate more social connections.

**Perceptions of social isolation.**   To assess the perception of social isolation, we used one item: "Do you consider yourself isolated in society?" This item was rated using a four-point scale ranging from 1 (*not at all*) to 4 (*very much*).

**Loneliness.**   We employed the three-item revised UCLA Loneliness Scale (R-UCLA) [45], which is a condensed form of the original 20-item revised UCLA Loneliness Scale [46]. Items were rated using a three-point scale ranging from 1 (*rarely*) to 3 (*often*) with high scores indicating high levels of loneliness.

**Depressive symptoms.**   The study used the Patient Health Questionnaire (PHQ-9) [47] to evaluate depressive symptoms. Nine items were rated using a four-point scale ranging from 0 (*not at all*) to 3 (*almost daily*). High scores indicate more severe depressive symptoms.

## Statistical analysis

We first conducted a chi-square test on the sociodemographic variables to ensure representativeness. We then conducted confirmatory factor analysis (CFA) to verify the structural validity of the LSNS-6. Based on previous research [42], we analyzed a model in which three items each represented the factors of the LSNS-6 Family and LSNS-6 Friend. Goodness of fit was assessed using several metrics: chi-square ($\chi^2$) test statistic, root mean squared error of approximation (RMSEA), comparative fit index (CFI), and the Tucker–Lewis index (TLI). A nonsignificant $\chi^2$ test statistics implies acceptable fit, but noting its sensitivity in large sample sizes is essential [48]. CFI and TLI values more than 0.95 indicate a good model fit [49], whereas RMSEA values less than 0.05 point to an optimal fit; values between 0.05 and 0.08 and between 0.08 and 0.10 denote moderate and marginal fit, respectively [50]. The study determined the reliability of each scale using Cronbach's alpha, a widely recognized measure of reliability [51]. An alpha value of 0.70 or higher is considered acceptable [52]. We then calculated the descriptive statistics for the LSNS-6 Family, LSNS-6 Friend, perception of social isolation, R-UCLA, and the PHQ-9 scale. Finally, we examined the relationships among these variables by computing Pearson's correlation coefficients.

We employed SEM to validate the model proposed in Fig 1. This process began with developing paths for each variable. We then refined the model by removing paths that were statistically nonsignificant. The significance of path coefficients was established at the 5% level. Path coefficients, denoted by $\beta$, indicate the strength of the relationship between variables (weak: < .20); moderate: .20 to .50); strong: >.50) [53]. The goodness of fit of the model was indicated by the $\chi^2$ test statistic, RMSEA, CFI, and TLI.

Given the large sample size, we considered the increased likelihood of obtaining statistically significant results for SEM, which necessitates caution in interpretation: the model validation involved data from respondents who provided complete and valid responses. The initial analysis included data from participants with unspecified or other gender identities for a comprehensive assessment of the model. We then conducted separate analyses for male and female participants to identify gender-specific differences within the model.

We performed mediation analysis using the bootstrap method to assess indirect effects within the SEM framework. This method takes a sample of researchers of size N and creates a new sample from which to extract the replacement N values of the independent, mediating, and dependent variables. For example, this option can be repeated 5,000 times to compute 5,000 estimates of indirect effects [54]. The bootstrap method is known for providing accurate confidence intervals for indirect effects, because this method is based on the empirical distribution of the estimates [55, 56]. We computed bootstrap confidence intervals corrected for bias using 5,000 resamples, which maintains 95% confidence interval. All statistical analyses were conducted using SPSS 28.0 and Amos 28.0.

## Results

### Characteristics of participants

The participants were composed of 1,805 (54.5%) men and 1,493 (45.0%) women, which represents a broad age range. Table 1 outlines the detailed sociodemographic characteristics of the respondents. To assess the representativeness of the sample relative to the Japanese population, we conducted chi-square tests for sex and age using demographic data from the Ministry of Internal Affairs and Communications as of March 2022 as reference. Notably, the sample consisted of fewer participants in their 20s and 70s than expected and more in their 40s and 50s compared with the broad Japanese population. This discrepancy poses the possibility that the data may only partially reflect the demographic composition of Japan.

### Validity of the LSNS-6 and reliability of each scale

We conducted CFA to test the structural validity of the LSNS-6. The model demonstrated a good fit ($\chi^2$ (8) = 81.03, $p < .001$, RMSEA = .05, CFI = .99, TLI = .99). This result confirmed the validity of the two-factor structure for the data. We reached Cronbach's alpha coefficients of 0.88 and 0.89 for LSNS-6 Family and LSNS-6 Friend, respectively, which indicates appropriate reliability for both scales. In addition, Cronbach's alpha coefficients for R-UCLA and PHQ-9 were 0.85 and 0.92, respectively, which were sufficiently reliable.

**Table 1. Characteristics of respondents.**

| Variable | N (%) | Variable | N (%) |
|---|---|---|---|
| Sex | | Occupation | |
| Male | 1,805 (54.5%) | Employed | 2,395 (72.2%) |
| Female | 1,493 (45.0%) | Homemaker | 392 (11.8%) |
| Other gender | 7 (0.2%) | Unemployed | 473 (14.3%) |
| Unknown | 10 (0.3%) | Student | 25 (0.8%) |
| Age group | | Other | 30 (0.9%) |
| 20–29 | 155 (4.7%) | Annual house hold income | |
| 30–39 | 409 (12.4%) | <2.0 million | 291 (8.7%) |
| 40–49 | 890 (26.8%) | <2.0–3.9 million | 533 (16.1%) |
| 50–59 | 1237 (37.4%) | <4.0–5.9 million | 566 (17.1%) |
| 60–69 | 549 (16.5%) | <6.0–7.9 million | 423 (12.8%) |
| 70– | 73 (2.1%) | ≥8.0 million | 694 (20.9%) |
| Unknown | 2 (0.1%) | Unknown | 808 (24.4%) |

## Descriptive statistics and correlation analysis

Table 2 presents the results of descriptive statistics and correlation analysis. Pearson's correlation coefficients indicated that LSNS-6 Family and LSNS-6 Friend were significantly and negatively correlated with perceived social isolation, loneliness, and depressive symptoms. Moreover, perceived social isolation was positively correlated with loneliness and depressive symptoms. Lastly, loneliness and depressive symptoms were significantly and positively correlated.

## Validation of the hypothetical model

We utilized SEM to evaluate the hypothetical model (Fig 1). Initially, we established paths between each variable as proposed in the model. We assumed a covariance between the LSNS-6 Family and LSNS-6 Friend error variables. The threshold for statistical significance of the path coefficients was set to 5%. Analysis revealed that the path that links social isolation from friends to depressive symptoms was statistically nonsignificant, which leads to its exclusion from the model. Fig 2 illustrates the model after omitting these nonsignificant paths. The study confirmed the goodness of fit of the model ($\chi^2(1) = 3.73$ (not significant); RMSEA = .03, CFI = 1.00, TLI = 1.00). The findings exhibited significant negative correlation of social isolation from family to perceived social isolation ($\beta = -.20$, 95% CI [−.24, −.17], $p < .001$) and of social isolation from friends to perceived social isolation ($\beta = -.24$, 95% CI [−.27, −.20], $p < .001$). Additionally, perceived social isolation displayed significant positive associations with loneliness ($\beta = .66$, 95% CI [.63, .68], $p < .001$) and depressive symptoms ($\beta = .27$, 95% CI [.23, .31], $p < .001$). Loneliness also exhibited a significant positive relationship with depressive symptoms ($\beta = .40$, 95% CI [.36, .44], $p < .001$). Furthermore, social isolation from family and friends exhibited significant negative paths to loneliness ($\beta = -.04$, 95% CI [−.06, −.01], $p < .01$; and $\beta = -.13$, 95% CI [−.15, −.10], $p < .001$, respectively). Moreover, social isolation from family demonstrated a notable negative correlation with depressive symptoms ($\beta = -.05$, 95% CI [−.08, −.02], $p < .001$). Given the sample size, the study inferred that social isolation from family and friends exerted a relatively minor impact on loneliness and depressive symptoms.

**Table 2. Descriptive statistics and correlations between variables.**

|   |   | Correlation coefficient ($r$) | | | | |
|---|---|---|---|---|---|---|
|   |   | 1 | 2 | 3 | 4 | 5 |
| 1 | LSNS-6 Family | – | | | | |
| 2 | LSNS-6 Friend | .43 *** | – | | | |
| 3 | Perception of social isolation | −.31 *** | −.32 *** | – | | |
| 4 | R-UCLA | −.29 *** | −.36 *** | .71 *** | – | |
| 5 | PHQ-9 | −.25 *** | −.23 *** | .57 *** | .61 *** | – |
| *Descriptive statistics* | | | | | | |
| | *Mean* | 5.10 | 3.28 | 2.21 | 5.03 | 5.53 |
| | *SD* | 3.33 | 3.49 | 0.92 | 1.94 | 6.24 |
| | *Minimum* | 0 | 0 | 1 | 3 | 0 |
| | *Maximum* | 15 | 15 | 4 | 9 | 27 |
| | *Skewness* | 0.14 | 0.85 | 0.31 | 0.65 | 1.40 |
| | *Kurtosis* | −.0.79 | −0.13 | −0.74 | −0.66 | 1.47 |

*Note*. LSNS-6 Family: Social isolation from family. LSNS-6 Friend: Social isolation from friends. Perception of social isolation: subjective perception of the degree of social isolation. R-UCLA: measures level of loneliness. PHQ-9: assesses the severity of depressive symptoms.

*** $p < .001$.

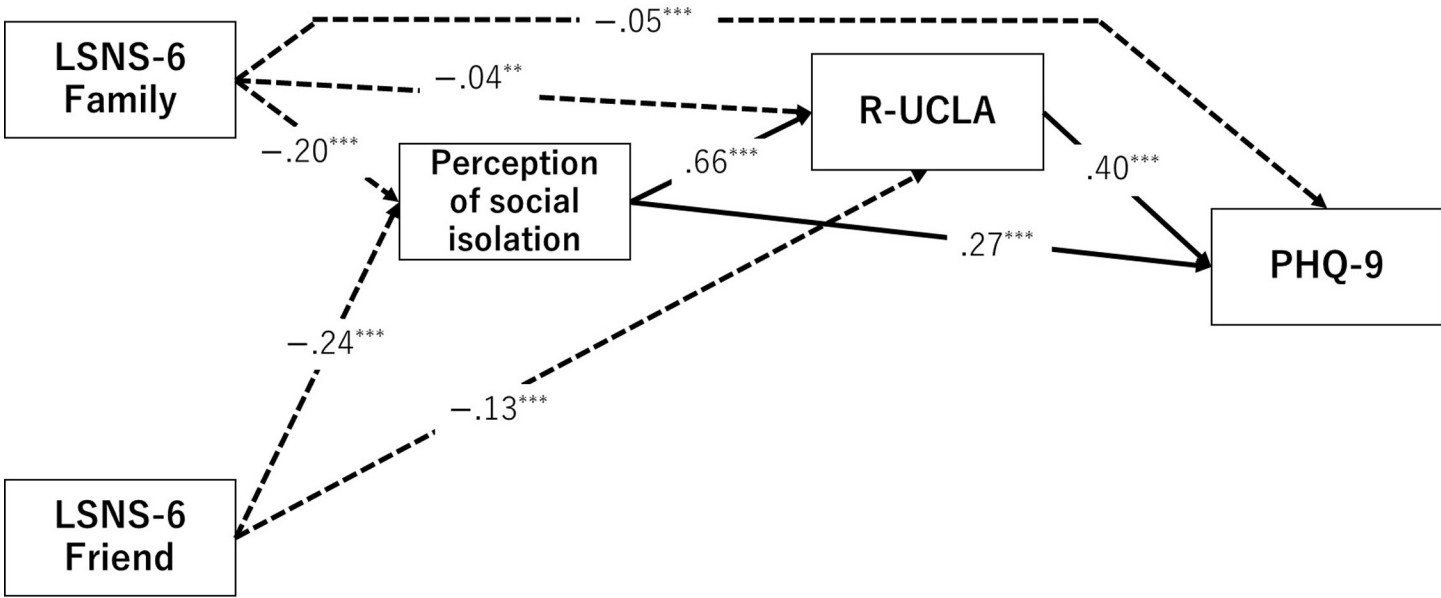

**Fig 2. Structural Equation Modeling (SEM) of the hypothetical model.** Rectangular shapes represent the psychological scales used in the study. Arrows indicate the nature of the relationships between these scales: dashed and solid arrows for negative and positive relations, respectively. For clarity, error variables and covariances were omitted from the illustration.

Table 3 presents the SEM results of the hypothetical model, which were categorized according to gender. The coefficients represent the paths identified through SEM analysis. For example, the values demonstrate that male and female participants exhibited a significant negative path from social isolation (family) to perceived social isolation, with minor differences in the

**Table 3.  Gender-specific results from structural equation modeling.**

|  |  | Men | | Women | |
|---|---|---|---|---|---|
|  | LSNS-6 Family→ PSI | −.21 | *** | −.22 | *** |
|  | LSNS-6 Friend→ PSI | −.21 | *** | −.27 | *** |
|  | LSNS-6 Family→ R-UCLA | − |  | −.08 | * |
|  | LSNS-6 Friend→ R-UCLA | −.13 | *** | −.14 | *** |
|  | LSNS-6 Family→ PHQ-9 | −.06 | *** | −.05 | *** |
|  | LSNS-6 Friend→ PHQ-9 | − |  | − |  |
|  | PSI→ R-UCLA | .67 | *** | .63 | *** |
|  | PSI→ PHQ-9 | .26 | *** | .27 | *** |
|  | R-UCLA→ PHQ-9 | .43 | *** | .38 | *** |
| *the goodness of fit of the model* |  |  |  |  |  |
|  | RMSEA | .02 | | .00 | |
|  | CFI | 1.00 | | 1.00 | |
|  | TLI | 1.00 | | 1.00 | |

*Note*. Values in the table represent the path coefficients derived from SEM analysis. PSI: perception of social isolation indicates one's perception of social isolation. LSNS-6 Family: Social isolation from family members. LSNS-6 Friend: Social isolation from friends. R-UCLA: measures the level of loneliness experienced by an individual. PHQ-9: assesses the severity of depressive symptoms.

$^{*}p < .05$

$^{***}p < .001.$

effects of variables between genders. Thus, we concluded that the results in Fig 2 apply to both genders.

The SEM results indicate that the perception of social isolation and loneliness could mediate the relationship between social networks and depressive symptoms. To investigate this possibility, we performed a mediation analysis using the bootstrap method with a focus on the indirect effects of the LSNS-6 Family on the perception of social isolation and loneliness leading to depressive symptoms. Analysis revealed significant indirect effects ($\beta = -.10$, 95% CI [$-.12, -.08$], $p < .001$), as evidenced by the 95% confidence interval excluding zero. Similarly, the impact of LSNS-6 Friend on the indirect relationship between social isolation and loneliness with depressive symptoms displayed a significant indirect effect ($\beta = -.11$, 95%CI [$-.13, -.09$], $p < .001$). These findings suggest that LSNS-6 Family and LSNS-6 Friend amplify loneliness and depressive symptoms through the mediation of perceived social isolation.

## Discussion

We explored the hypothetical model illustrated in Fig 2, which displays the influence of social networks, social isolation, and loneliness on depressive symptoms. The findings revealed that the direct effect of the size of social networks on loneliness and depressive symptoms was marginal. However, comprehensively analyzing social networks, perception of social isolation, and loneliness and examining their effects on depressive symptoms identified a mechanism in which the relationship between social network size and depressive symptoms was mediated by perception of social isolation and loneliness, which increase feelings of loneliness and depressive symptoms. This result extends those of previous studies that examined these variables in isolation [24–26, 29–32].

The results of the current study and prior research that primarily focus on adolescents [25] underscore the age-independent trend of this finding. Moreover, the gender-based SEM of the proposed model identified no significant disparities between men and women. Thus, the current results indicate that perception of social isolation and loneliness are critical mediators that can worsen mental health challenges regardless of gender. Although the majority of studies and intervention methods on social isolation and loneliness have focused on older adults [18, 35, 57], the findings are significant for demonstrating that social isolation and loneliness can lead to a decline in mental health regardless of age or gender, which underscores their universal applicability.

Interestingly, the mechanism identified in this study suggests that perception of isolation increases loneliness instead of the actual state of social isolation, which was assessed using the sheer number of social networks as a proxy. This finding implies that merely expanding social networks may be unable to effectively diminish loneliness due to perceived social isolation [25]. The size of social networks exerts a minimal direct impact on depressive symptoms, whereas the perception of social isolation and feelings of loneliness play more direct roles in contributing to these symptoms. Notably, loneliness serves as a mediator, which indicates that increased feelings of loneliness and perceptions of social isolation could intensify depressive symptoms.

The findings underscore the importance of addressing the transition from perception of social isolation and loneliness to depressive symptoms. Interventions that target altering the perception of social isolation can help reduce depressive symptoms and alleviate loneliness if they target cognitive and emotional aspects across age groups. Psychoeducational approaches and cognitive–behavioral therapy for changing cognition and emotion effectively alleviate loneliness [34]. These interventions can be pivotal in transforming perceptions and reducing loneliness and depressive symptoms. Additionally, reminiscence-based psychological interventions

have demonstrated effectiveness [58]. We advocate for strategies that focus on the cognitive and emotional facets of social isolation and loneliness instead of the mere expansion of social networks [37], because social isolation does not necessarily lead to loneliness [23].

Previous research underscores the importance of primary prevention in addressing social isolation and loneliness [59–61]. However, studies on the effectiveness of such preventive measures are limited, which emphasizes the need to develop intervention programs that focus on primary prevention against social isolation and loneliness. The study suggests the development of early psychoeducational programs for school education and community-based cognitive–behavioral therapeutic interventions to prevent social isolation and loneliness. Such initiatives could play a vital role in preventing depressive symptoms emerging from social isolation and loneliness.

The following limitations constrain the findings. First, the study was unable to establish causal relationships due to its cross-sectional nature. Future research using longitudinal data would be beneficial for understanding the evolution of perception of social isolation and its consequent impact on loneliness and depressive symptoms. Second, reliance on self-reported data could introduce bias, as individuals who are socially isolated or acutely lonely may negatively respond due to inherent negative self-perceptions [30]. Future studies should incorporate latent markers and third-party evaluations alongside current methods. Lastly, the online nature of data collection may lead to sampling bias. Data may not fully represent the Japanese population; notably, the survey may have missed individuals without Internet access or with severe social isolation or loneliness. To enhance these findings, further research should consider conducting in-depth interviews with people that encounter social isolation or intense loneliness (e.g., individuals who have withdrawn from society) and demographic studies in specific communities.

Despite these challenges, the significance of this study lies in its extensive, data-driven exploration of the contribution of social isolation, perception of social isolation, and loneliness to depressive symptoms. The identified patterns are essential for developing support strategies for individuals facing social isolation and loneliness. Future research should build on these insights and focus on specific subgroups, including those who are socially isolated, and use various methods of evaluation across diverse cohorts.

## Conclusion

This study provides an in-depth analysis of the interplay among social networks, perception of social isolation, and loneliness, and their collective impact on depressive symptoms. Although social isolation and loneliness were long associated with depressive symptoms, the findings reveal that the individual perception of isolation and the consequent feelings of loneliness play more crucial roles in the development of depressive symptoms instead of the actual state of social isolation. This finding highlights the importance of not only expanding social networks but also addressing and correcting perceptions of social isolation and loneliness as critical strategies for preventing depressive symptoms.

## Supporting information

**S1 Dataset. Anonymized data set.**
(XLSX)

**S1 Checklist. STROBE-checklist-v4-combined-PlosMedicine.**
(DOCX)

## Author Contributions

**Conceptualization:** Miyuki Aiba, Haruhiko Midorikawa, Hirokazu Tachikawa.

**Data curation:** Natsuho Kushibiki, Miyuki Aiba.

**Formal analysis:** Natsuho Kushibiki.

**Funding acquisition:** Hirokazu Tachikawa.

**Investigation:** Miyuki Aiba, Kentaro Komura.

**Methodology:** Miyuki Aiba.

**Project administration:** Takafumi Ogawa, Chie Yaguchi, Hirokazu Tachikawa.

**Supervision:** Daichi Sugawara, Yuki Shiratori, Naoaki Kawakami.

**Writing – original draft:** Natsuho Kushibiki.

**Writing – review & editing:** Miyuki Aiba, Daichi Sugawara, Hirokazu Tachikawa.

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
