## [Decision Letter · Decision Letter 0]

11 Jul 2023

PONE-D-23-09807How can we reduce the effects of social isolation and loneliness on depression? Findings from an online survey of  a Japanese populationPLOS ONE

Dear Dr. Tachikawa,

Thank you for submitting your manuscript to PLOS ONE. After careful consideration, we feel that it has merit but does not fully meet PLOS ONE’s publication criteria as it currently stands. Therefore, we invite you to submit a revised version of the manuscript that addresses the points raised during the review process. Thank you for submitting your work to PLOS. The reviewers both expressed some concerns with the study as reported. Please attend to Reviewer 1's suggestion for more consistent and accurate language. Terms should be consistent with the assessed constructs.

Reviewer 2 commented on the lack of a specific population that your findings may be generalized to. Please clarify whether your study was aimed at generalizing associations between specific factors, or identifying prevalence rates and risk ratios of a given population. You might consider some reporting guidelines, such as STROBE for cross-sectional designs. Some information that could be of interest is, did your study include the full assessed range of your variables, covering lowest to highest scores? Testing and controlling for demographic confounds may also contribute to our understanding of the sample and variables of interest. Please respond to the terminology and accuracy concerns, as well as whether the sample represents a specific population or model testing of associations between select variables.

We look forward to receiving your revised manuscript.

Kind regards,

Keith M. Harris, PhD

Academic Editor

PLOS ONE

Journal Requirements:

“This work was supported by JST RISTEX “SOLVE for SDGs: Preventing Social Isolation & Loneliness and Creating Diversified Social Networks” Grant Number JPMJRX21K2, Japan.”

“This work was supported by JST RISTEX “SOLVE for SDGs: Preventing Social Isolation & 303 Loneliness and Creating Diversified Social Networks” Grant Number JPMJRX21K2, Japan.”

“This work was supported by JST RISTEX “SOLVE for SDGs: Preventing Social Isolation & Loneliness and Creating Diversified Social Networks” Grant Number JPMJRX21K2, Japan.”

Reviewers' comments:

Reviewer's Responses to Questions

**Comments to the Author**

1. Is the manuscript technically sound, and do the data support the conclusions?

Reviewer #1: Partly

Reviewer #2: Partly

2. Has the statistical analysis been performed appropriately and rigorously? 

Reviewer #1: Yes

Reviewer #2: Yes

3. Have the authors made all data underlying the findings in their manuscript fully available?

Reviewer #1: Yes

Reviewer #2: Yes

4. Is the manuscript presented in an intelligible fashion and written in standard English?

Reviewer #1: Yes

Reviewer #2: Yes

5. Review Comments to the Author

Reviewer #1: ID: PONE-D-23-09807

Title: How can we reduce the effects of social isolation and loneliness on depression?

Findings from an online survey of a Japanese population

Thank you for providing a chance to review this manuscript.

Comment: Major Revision.

Detailed information:

Abstract

Objective: Need a simple sentence to summarize the background and why this research is being conducted.

Methods：What kind of model is built using structural equations.

Conclusions: How did the new term 'social support' suddenly appear in the conclusion section, which seems abrupt? Please present important conclusions based on the research purpose and results.

Introduction

Page 3, Line 80: A very low-level error occurred.The first letter of a word after a colon does not need to be capitalized. If you want to express the former, it's best not to use 'first' to avoid making people think it's the first difference.

Page 3, Line 83: Why use "Additionally" to connect the two sentences before and after? The expression is incorrect because you are explaining the "effect" in the previous sentence.

Page 3, Line 89-95: Why is the first sentence in this paragraph contradictory to the following content? Please pay attention to logic and language expression.

Page 4, Line 111-117: As an outcome variable, the explanation of depression is too brief. Please add relevant introductions.

Page 4, Line 125: A brief explanation of the content of Figure 1 needs to be added here.

Overall: The introduction section on social isolation and loneliness is verbose and lacks logic, please refine it.

Methods

Page 6, Participants and Procedures: What is your sample collection method? How do you consider the size of the sample size, and how do you handle samples with unknown gender or 'other'?

Page 6, Measures , Line 155-157: It should be the data result indicating that the internal consistency is good. Please express the semantics more rigorously. Also, what does A refer to? The full name should be written out. Finally, reference literature is needed to prove the source and credibility of this value.

Page 6, Measures , Line 164: Same problem as above.

Page 6, Measures , Line 168: Same problem as above.

Results

Page 4, Line 206-208: So what variables are the results of the values in Table 2? Please describe clearly and write it down in Table 2.

Page 4, Line 209-212: This paragraph should be explained in the statistical analysis section of the methods section.

Overall: For the convenience of readers, explanations for variable names in the table can be added in the comments below, such as "LSNS-6 Family" and "R-UCLA". It is necessary to explain what they represent again.

In general, 1) Please highlight important information. 2) Strengthen logic. The reader's thinking should be guided by you, rather than having the reader struggle to clarify your logic on their own. 3) The English expression of the entire text is problematic and needs to be repaired. 4) The differences between various independent variables need to be clearer to avoid confusion.

Thank you and my best,

Your reviewer

Reviewer #2: It is my understanding that this study is the result of an analysis of how social networks and the perception of social isolation and loneliness affect depression among participants of an Internet survey in Japan.

Clarifying the relationship of social isolation and loneliness in the structure of depression would be an important factor to consider in depression prevention measures.

However, this study has serious problems that cannot be scientifically ignored in order to be published in an international journal.

The biggest problem is that this study uses a sample sampled from an unspecified population using the Internet. While Internet-based surveys have the advantage of being less costly and labor intensive, they have the serious problem that the representativeness of the sample is unknown. There is likely to be a strong bias to participate in survey monitoring, and there may be considerable psychological bias. For conducting research on depression and other mental illnesses, Internet-based surveys are difficult to estimate bias, and it would be difficult to seek generality in their results.

Therefore, this study is considered unsuitable for publication in an international journal because the possibility of scientific inaccuracy cannot be denied.

Publication in a domestic journal would enhance its value.

6. PLOS authors have the option to publish the peer review history of their article (what does this mean?). If published, this will include your full peer review and any attached files.

Reviewer #1: No

Reviewer #2: No

---

## [Author Response · Author response to Decision Letter 0]

8 Sep 2023

Response to academic editor and reviewers

Journal Requirements:

Response:

We have now revised the PLOS ONE’s style requirement, to ensure that the new version of the manuscripts fulfils these requirements.

“This work was supported by JST RISTEX “SOLVE for SDGs: Preventing Social Isolation & Loneliness and Creating Diversified Social Networks” Grant Number JPMJRX21K2, Japan.”

Response:

We have described the role of the funders below.

“This work was supported by JST RISTEX “SOLVE for SDGs: Preventing Social Isolation & Loneliness and Creating Diversified Social Networks” Grant Number JPMJRX21K2, Japan. The funders had no role in study design, data collection and analysis, decision to publish, or preparation of the manuscript.”

“This work was supported by JST RISTEX “SOLVE for SDGs: Preventing Social Isolation & 303 Loneliness and Creating Diversified Social Networks” Grant Number JPMJRX21K2, Japan.”

“This work was supported by JST RISTEX “SOLVE for SDGs: Preventing Social Isolation & Loneliness and Creating Diversified Social Networks” Grant Number JPMJRX21K2, Japan.”

Response:

Thank you for pointing this out. We have removed the text regarding funding from the manuscript and revised the Funding Statement below.

“This work was supported by JST RISTEX “SOLVE for SDGs: Preventing Social Isolation & Loneliness and Creating Diversified Social Networks” Grant Number JPMJRX21K2, Japan. The funders had no role in study design, data collection and analysis, decision to publish, or preparation of the manuscript.”

Response:

Our data use statement remains unchanged. We understand that the manuscript will be held until the accession number or DOI required to access the data is provided.

Response:

We appreciate your pointing this out. The Supporting Information file caption is included at the end of the manuscript.

See the next page for responses to Reviewer 1.

Responses to Reviewer #1:

Thank you for providing us with an opportunity to revise this manuscript.

Comment: Major Revision.

Detailed information:

Response:

Thank you for providing us with your insights. We appreciate the opportunity to revise the manuscript in detail. We have modified the entire manuscript as you suggested. Based on your comments, we have made the following revisions.

Abstract

Objective: Need a simple sentence to summarize the background and why this research is being conducted.

Response:

Thank you for your comment. We have included a sentence summarizing the research background and why we conducted this research, as shown below. 

“Understanding the interplay between social isolation, loneliness, and the perception of social isolation on depressive symptoms is crucial for effective intervention planning” (pp1, line 3-4).

Methods: What kind of model is built using structural equations.

Response:

Thank you for your comment. We have incorporated your comments as shown below.

“We assessed the participants’ social network size using the Lubben Social Network Scale, perceived social isolation with a 1-item question, loneliness with the 3-item UCLA Loneliness Scale, and depressive symptoms with the Patient Health Questionnaire-9. Structural equation modeling employed to test the hypothesized model for the effects of social networks, the perception of social isolation, and loneliness on depressive symptoms.” (pp1, lines 9-13)

Conclusions: How did the new term’ social support’ suddenly appear in the conclusion section, which seems abrupt? Please present important conclusions based on the research purpose and results.

Response:

Thank you for your comment. We agree with your comment and have changed the term from ‘social support’ to ‘social network,’ as shown below. 

“It is vital to consider both the quality of social networks and the nature of individual relationships in addition to expanding a person’s social network.” (pp1, line 22-23)

Introduction

Page 3, Line 80: A very low-level error occurred. The first letter of a word after a colon does not need to be capitalized. If you want to express the former, it’s best not to use ‘first’ to avoid making people think it’s the first difference.

Response:

We thank you for the careful review. While reviewing the text, we removed this sentence.

Page 3, Line 83: Why use “Additionally” to connect the two sentences before and after? The expression is incorrect because you are explaining the “effect” in the previous sentence.

Response:

Thank you for your comment. We agree with you and removed the word ‘Additionally’

Page 3, Line 89-95: Why is the first sentence in this paragraph contradictory to the following content? Please pay attention to logic and language expression.

Response:

Thank you for your comment. The passage you pointed out was intended to argue that the correlation between social isolation and loneliness is weak and that social isolation and loneliness do not necessarily co-occur. To clarify our point, we have rewritten the passage paying attention to logic and language expressions as shown below.

“Notably, previous research has shown only a weak or modest correlation between social isolation and loneliness [14], indicating that having many objective social connections do not inherently negate feelings of loneliness [15]. Therefore, viewing social isolation and loneliness as conceptually distinct constructs and independently assessing them is critical [6].” (pp2, line 47-51)

Page 4, Line 111-117: As an outcome variable, the explanation of depression is too brief. Please add relevant introductions.

Response:

Thank you for your suggestion. We have added an introduction to previous studies to further emphasize that depressive symptoms are a significant outcome of problems such as social isolation and loneliness, as shown below. 

“Social isolation, acting as a direct stressor, elevates brain stress responses [12], and long-standing depression has been partially explained by social isolation [18]. Prior research indicates that indicators of social isolation leading to depression encompass living alone, possessing a fragile social network, and sparse social interaction [19,20]. Traditionally, loneliness has been intrinsically linked with depression [21], even surpassing objective metrics of social connectedness as a predictive factor for depression [22, 23]. Irrespective of age, profound loneliness associated with severe depressive symptoms [24,25]. Moreover, the perceived lack of societal support and social isolation can lead to depressive symptoms, hindering depressed patients’ recovery [26]. Thus, while many studies have highlighted the relationship between social isolation, loneliness, and perception of social isolation, limited research has concurrently treated these elements as discrete concepts and investigated their serial influence on depression.” (pp3, line 60-70)

Page 4, Line 125: A brief explanation of the content of Figure 1 needs to be added here.

Response:

Thank you for your comment. We have briefly explained the model in Figure 1, as shown below.

“Based on antecedent research, we formulated a hypothetical (Fig 1) to illuminate the effect of social isolation (quantified by social network size), its perceptions, and loneliness on depressive symptoms. Table 1 shows a concise summary of these characteristics. Our proposed model postulates that the size of social networks, perceptions of social isolation, and loneliness all converge to influence depressive symptoms. Additionally, perceptions of social isolation and loneliness potentially mediate the connection between social networks and depressive symptoms.” (pp3, line 77-78, pp4, line 79-85)

Overall: The introduction section on social isolation and loneliness is verbose and lacks logic, please refine it.

Response:

Thank you for your comment. We reviewed the logical development of the rest of the Introduction and completely revised the Introduction based on your suggestion.

Methods

Page 6, Participants and Procedures: What is your sample collection method? How do you consider the size of the sample size, and how do you handle samples with unknown gender or ‘other’?

Response:

Thank you for your comment. We have included specific description of the sample collection method. 

“This study was conducted in collaboration with the online research company, Cross Marketing Inc., spanning the period of March 23 to 28, 2022. Cross Marketing Inc. Participants were recruited through a website from the 3.55 million monitor base of Cross Marketing Inc., targeting Japanese individuals aged 20 and above. This recruitment procedure ensured targeting a broad spectrum of respondents throughout Japan. The survey was completed when we received responses from 3,500 participants.” (pp5, line 95-99)

The sample size of this study is larger than in other studies using structural equation modeling. We obtained as large a sample as possible to examine trends across Japan. However, e large samples make significant results more likely. Therefore, caution should be exercised in interpreting the results. We have mentioned this in the manuscript, as shown below.

“We specifically sought a larger sample size to identify trends in pertinent to the Japanese population.” (pp5, line 100)

“Given the large sample size, there is an increased likelihood of obtaining statistically significant results when employing structural equation modeling, hence requiring careful interpretation of findings.” (pp6, line 1844, pp7, line 145-146)

Moreover, as mentioned earlier, this study aimed to examine trends in Japan as a whole. Therefore, we conducted the analysis without excluding participants with unknown gender or other gender participants. We described the treatment of unknown gender and others, as indicated below.

“In the initial phase, for a comprehensive assessment of the hypothetical model, the analysis incorporated data from participants with gender identified as unknown or other.” (pp7, line 147-148)

Page 6, Measures, Line 155-157: It should be the data result indicating that the internal consistency is good. Please express the semantics more rigorously. Also, what does A refer to? The full name should be written out. Finally, reference literature is needed to prove the source and credibility of this value.

Page 6, Measures, Line 164: Same problem as above.

Page 6, Measures, Line 168: Same problem as above.

Response:

We appreciate these helpful suggestions. A specific explanation of Cronbach’s alpha, an indicator of the scale's reliability, was missing in the original version of the manuscript. Therefore, we have added a description of Cronbach’s alpha and the references, as shown below.

“We evaluated the reliability of each scale by determining Cronbach's alpha, a widely recognized measure of reliability [36]. An alpha value of 0.70 or higher is deemed acceptable [37].” (pp6, line 132-134)

Thank you for your comment. As shown below, we also included Cronbach’s alpha ratings for each scale in the results to facilitate the reader’s understanding.

“Cronbach's alphas for of the scales were adequate because they were greater than .70.” (pp7, line 158-159)

Results

Page 4, Line 206-208: So what variables are the results of the values in Table 2? Please describe clearly and write it down in Table 2.

Response:

Thank you for your comment. We appreciate this suggestion. Table 2 has been renumbered from Table 2 to Table 3 to reflect other additions. Table 3 shows the values of the path coefficients obtained for the hypothetical model by gender from structural equation modeling. We have described the meaning of Table 3 and how to read the table, as shown below.

“The coefficients in Table 3 represent the paths derived from the structural equation modeling. As an illustration, the values in Table 3 suggest that both male and female participants exhibited a significant negative path from family social isolation to perceived social isolation.” (pp9, line 193-196)

Page 4, Line 209-212: This paragraph should be explained in the statistical analysis section of the methods section.

Response:

I agree with your opinion. We have removed the paragraph from the relevant section and included the text in the Statistical Analysis section, as shown below.

“We conducted mediation analysis using the Bootstrap method to quantify indirect effects within the SEM model. Calculating the bias-corrected Bootstrap confidence intervals involved 5,000 Bootstrap resamples (with a 95% confidence interval).” (pp7, line 151-153)

Overall: For the convenience of readers, explanations for variable names in the table can be added in the comments below, such as “LSNS-6 Family” and “R-UCLA.” It is necessary to explain what they represent again.

Response:

Thank you for your suggestions. We have described each variable under Tables 2 and 3, as shown below.

“LSNS-6 Family; Social isolation from the family, LSNS-6 Friend; Social isolation from friends, Perception of Social Isolation; An individual’s perceptions of their social isolation degree, R-UCLA; Level of loneliness, PHQ-9; Level of depressive symptoms” (pp8, line 172-174, pp10, line 216-218)

In general, 1) Please highlight important information. 2) Strengthen logic. The reader’s thinking should be guided by you, rather than having the reader struggle to clarify your logic on their own. 3) The English expression of the entire text is problematic and needs to be repaired. 4) The differences between various independent variables need to be clearer to avoid confusion.

Response:

Thank you for taking the time to review our manuscript. We have revised the overall structure of the discussion and highlighted important information. In addition, the structure of the Introduction and Discussion was modified to strengthen the logic. We reviewed and revised the English expressions throughout the text. We have also included Table 1, summarizing the terms at the end of the introduction to clarify the differences among the independent variables. Additionally, for consistency, we changed the title depression to depressive symptoms. In addition, the variable names in Fig. 1 have been changed to avoid confusion for the reader.

Table 1. Characteristics of social isolation, perceptions of social isolation, and loneliness.

Variable Definition and Characteristics

Social isolation an objective state of having few social contacts with others.

A typical indicator of social isolation is the network size.

Perception of social isolation an individual's perception of the extent of his or her social isolation.

It is an individual's cognitive evaluation, which is different from the number of networks.

Loneliness The lack of social connections (social loneliness) or negative feelings (emotional loneliness) that arise when the quantity or quality of relationships with a particular partner or peer is subjectively deficient compared to the person's ideal

(pp4, line 87-88)

TITLE: “How can we reduce the effects of social isolation and loneliness on depressive symptoms? Findings from an online survey of a Japanese population”

Fig 1. Hypothetical model of this study. The squares indicate a psychological variable, and the arrows indicate associations among variables.

Responses to Reviewer #2:

It is my understanding that this study is the result of an analysis of how social networks and the perception of social isolation and loneliness affect depression among participants of an Internet survey in Japan.

Clarifying the relationship of social isolation and loneliness in the structure of depression would be an important factor to consider in depression prevention measures.

However, this study has serious problems that cannot be scientifically ignored in order to be published in an international journal.

The biggest problem is that this study uses a sample sampled from an unspecified population using the Internet. While Internet-based surveys have the advantage of being less costly and labor intensive, they have the serious problem that the representativeness of the sample is unknown. There is likely to be a strong bias to participate in survey monitoring, and there may be considerable psychological bias. For conducting research on depression and other mental illnesses, Internet-based surveys are difficult to estimate bias, and it would be difficult to seek generality in their results.

Therefore, this study is considered unsuitable for publication in an international journal because the possibility of scientific inaccuracy cannot be denied.

Publication in a domestic journal would enhance its value.

Response:

Thank you for providing these insights. Your point is correct, but many studies published in PLOS ONE use structural equation modeling of online surveys (Fischer et al., 2014; Soon et al., 2020; Aljaberi et al., 2022). In addition, our study aimed to understand the associations and mechanisms of different variables and not to determine the prevalence or risk ratio in a given population, which we have clearly stated, as shown below. We considered the title to be misleading and have revised it.

“Consequently, this study aims to clarify the interrelations and underlying mechanisms connecting social isolation, perceptions of social isolation, loneliness, and depressive symptoms.” (pp3, line 75-76)

Also, as you point out, based on STROBE regarding cross-sectional studies, we recognize the potential biases and imprecise representativeness inherent in data of cross-sectional designs, which limits the validity of cross-sectional studies. Therefore, we reviewed the reporting guidelines for cross-sectional designs and described sampling and data representativeness as limitations of this study, as described below. 

“Lastly, our online data collection method could induce sampling bias, potentially favoring those with ready internet access or those particularly concerned about social isolation and loneliness.” (pp12, line 265-267)

In addition, we have added the maximum, minimum, kurtosis, and skewness of each variable in Table 2, providing additional information about our sample. We hope that these revisions, based on your valuable feedback will improve the quality of this paper.

Table 2. Descriptive statistics and correlations between the variables.

　 　 　 1 2 3 4 5

1 LSNS-6 Family ― 

2 LSNS-6 Friend .43 *** ― 

3 Perception of social isolation -.31 *** -.32 *** ― 

4 R-UCLA -.29 *** -.36 *** .71 *** ― 

5 PHQ-9 -.25 *** -.23 *** .57 *** .61 *** ―

Descriptive statistics 

 Mean 5.10 3.28 2.21 5.03 5.53

 SD 3.33 3.49 0.92 1.94 6.24

 Minimum 0 0 1 3 0

 Maximum 15 15 4 9 27

 Skewness 0.14 0.85 0.31 0.65 1.40

 Kurtosis -0.79 -0.13 -0.74 -0.66 1.47

　 Cronbach's αlpha .88 .89 ― .85 .92

(pp8, line 171-174)

References

Fischer S, Lemmer G, Gollwittzer M, Nater UM (2014) Stress and Resilience in Functional Somatic Syndromes – A Structural Equation Modeling Approach. PLoS ONE 9(11): e111214. https://doi.org/10.1371/journal.pone.0111214

Soon JM, Wahab IRA, Hamdan RH, Jamaludin MH (2020) Structural equation modelling of food safety knowledge, attitude and practices among consumers in Malaysia. PLoS ONE 15(7): e0235870. https://doi.org/10.1371/journal.pone.0235870

Aljaberi MA, Alareqe NA, Alsalahi A, Qasem MA, Noman S, Uzir MUH, et al. (2022) A cross-sectional study on the impact of the COVID-19 pandemic on psychological outcomes: Multiple indicators and multiple causes modeling. PLoS ONE 17(11): e0277368. https://doi.org/10.1371/journal.pone.0277368

---

## [Decision Letter · Decision Letter 1]

23 Oct 2023

PONE-D-23-09807R1How can we reduce the effects of social isolation and loneliness on depressive symptoms? Findings from an online survey of  a Japanese populationPLOS ONE

Dear Dr. Tachikawa,

Thank you for submitting your manuscript to PLOS ONE. After careful consideration, we feel that it has merit but does not fully meet PLOS ONE’s publication criteria as it currently stands. Therefore, we invite you to submit a revised version of the manuscript that addresses the points raised during the review process.

You made some helpful revisions and good progress on this work. There are still, however, some fairly significant issues with lack of details of some methods, and statements that don't always fit with the study. Please pay particular attention to the requests from both reviewers for more details on the methodology. In addition, both reviewers mentioned statements in the manuscript that don't quite match the study content. That relates to criteria 4 here: https://journals.plos.org/plosone/s/criteria-for-publication

We look forward to receiving your revised manuscript.

Kind regards,

Keith M. Harris, PhD

Academic Editor

PLOS ONE

Reviewers' comments:

Reviewer's Responses to Questions

**Comments to the Author**

1. If the authors have adequately addressed your comments raised in a previous round of review and you feel that this manuscript is now acceptable for publication, you may indicate that here to bypass the “Comments to the Author” section, enter your conflict of interest statement in the “Confidential to Editor” section, and submit your "Accept" recommendation.

Reviewer #1: (No Response)

Reviewer #3: All comments have been addressed

2. Is the manuscript technically sound, and do the data support the conclusions?

Reviewer #1: Partly

Reviewer #3: Yes

3. Has the statistical analysis been performed appropriately and rigorously? 

Reviewer #1: Yes

Reviewer #3: Yes

4. Have the authors made all data underlying the findings in their manuscript fully available?

Reviewer #1: Yes

Reviewer #3: Yes

5. Is the manuscript presented in an intelligible fashion and written in standard English?

Reviewer #1: No

Reviewer #3: Yes

6. Review Comments to the Author

Reviewer #1: ID: PONE-D-23-09807R1

Title: How can we reduce the effects of social isolation and loneliness on depressive symptoms? Findings from an online survey of a Japanese population

Thank you for providing a chance to review this manuscript.

Detailed information:

Abstract

Methods：How do you determine that the surveyed sample is a “representative sample”? Is it sufficiently representative? Can it represent the population of Japan?

Conclusions: What are the intervention measures? Please write the key conclusion based on your question: “How can we reduce the effects of social isolation and loneliness on compressive symptoms”.

Methods

1.Participants and Procedures: You still haven't solved the core problem. 1) In the research design stage, the size of the sample size should be calculated. Secondly, how is the study population determined? Why choose this research population? 3) Didn't even collect demographic information?

2.Measures, Lines 112-117: You have not made any modifications yet. You should add data that can demonstrate the reliability and validity of the scale, and this data should have references.

In general, there are still issues with this study, as your population is unclear and confounding factors have not been considered.

Thank you and my best,

Your reviewer

Reviewer #3: I appreciate the opportunity to review the manuscript titled "How can we reduce the effects of social isolation and loneliness on depressive symptoms: Findings from an online survey of a Japanese population." This study investigates the correlation between social isolation, loneliness, and perceived social isolation in connection with depressive symptoms in the adult Japanese population. The authors conducted an online survey among 3315 representative Japanese adults to investigate these connections. The survey incorporated the Lubben Social Network Scale to gauge the size of the participants' social networks, the UCLA Loneliness Scale consisting of 3 items to evaluate loneliness, and the Patient Health Questionnaire-9 to measure depressive symptoms. In particular, the extensive sample size of 3315 participants enhances the robustness of the findings. The authors employed structural equation modelling to examine their hypothetical model, which considered the impact of social networks, perception of social isolation, and loneliness on depressive symptoms. The results revealed that larger social networks exhibited a weak correlation with lower levels of loneliness and depressive symptoms. Specifically, the study indicated that perceived social isolation was associated with increased levels of both loneliness and depressive symptoms.

In general, this study has considerable merit. To further enrich the scientific value of the manuscript, I have outlined some key points for the authors' consideration based on the feedback from the two reviewers. My aim is to ensure a comprehensive understanding of the comments of the reviewers, to evaluate the adequacy of the authors' responses, and to provide additional suggestions to enhance the manuscript.

TITLE

1. While the current title of the study provides some information on its focus, I would recommend a more descriptive one such as "Examining the Impact of Social Networks and Perceived Social Isolation on Depressive Symptoms among Adults in Japan". At its current stage, it would be premature to conclude that the study aims to mitigate the negative impact of social isolation and loneliness on depressive symptoms.

ABSTRACT

The abstract provides a comprehensive overview of the study. However, a few improvements could enhance its clarity and impact.

2. Please consider changing (“We assessed the size of the social network size using the Lubben Social Network Scale”) to (“We used the Lubben Social Network Scale to determine the size of the social network”).

3. The statistical significance of the goodness of fit of the model should be emphasised. Providing a brief interpretation of the beta coefficients and their implications for the interaction between social isolation, loneliness, and depressive symptoms would make the results more accessible to a wider audience.

4. The conclusion could be strengthened by reiterating the practical implications of the study findings and how they can contribute to the development of effective intervention strategies. Furthermore, providing some suggestions for future research directions can add depth to the conclusion.

INTRODUCTION

5. Although the study's hypothetical model is briefly mentioned, providing a more detailed integration of the model within the context of the existing literature could strengthen the background and set the stage for the study's aims and objectives.

6. I suggest narrating the terms in the text and providing relevant literature instead of intruding on a table to define variables, as it breaks the conventional approach.

7. The relevance of this entry is not clear (Fig 1. Hypothetical model of this study. The squares indicate a psychological variable and the 1 arrows indicate associations among variables.)

8. The authors did mention some of the expressions of loneliness, including hikikomori. There is an additional concept that correlates with loneliness that needs to be mentioned: social avoidant personality disorder, Taijin kyofusho, Modern Type Depression, Unemployment, and Not in Employment Education or Training (NEET). The economic and demographic factors that contribute to increased loneliness in Japan need to be further elaborated.

METHOD

9. In the abstract, we were told 'Lubben Social Network Scale… with a 1-item question' and in the method, we were told (“The LSNS-6 comprises six items…”.). Please reconcile.

10. The authors have described the outcome measures under the subheading "Measures". However, it may be necessary to mention the psychometric properties of the Japanese version of these measures. On the other hand, in line 131, pg. 6, it was stated that “Initially, we computed descriptive statistics for LSNS-6 Family, LSNS-6 Friend, Perception of Social Isolation, R-UCLA, and PHQ-9. We evaluated the reliability of each scale by determining Cronbach's alpha, a widely recognized measure of reliability [36]. An alpha value of 0.70 or higher is deemed acceptable [37]”. Exercises to explore reliability should be detailed in the subheading ‘Measures’ and removed from the subheading ‘statistical analysis’. Some of the literature cited appears to allude that the Japanese reliability versions of these instruments. Indeed, the authors did cite the relevant study but did not narrate the issue of the Japanese version of The LSNS-6 (Kurimoto A, Awata S, Ohkubo T, Tsubota-Utsugi M, Asayama K, Takahashi K, Suenaga K, Satoh H, Imai Y. [Reliability and validity of the Japanese version of the abbreviated Lubben Social Network Scale]. Nihon Ronen Igakkai Zasshi. 2011;48(2):149-57. Japanese. doi: 10.3143/geriatrics.48.149.

11. Although the method states that efforts were made to ensure diversity in the participant pool, it would be beneficial to provide more explicit details on the specific measures taken to ensure representation across various sociodemographic factors, such as age, location (rural and urban), socioeconomic status, and cultural background. Including a breakdown of these demographics in the participant pool can add credibility to the study's results.

12. Although the method briefly mentions that participants were informed about the study objectives and ethical considerations, it would be advantageous to provide more detailed information on the informed consent process. Specifically, explaining how participants were informed about their rights, the purpose of the study, the possible risks and benefits, and how their data would be used and protected can help ensure transparency and build trust with participants.

13. Although the method mentions that participants were given reward points as a token of appreciation, it might be useful to elaborate on the nature of these incentives and how they were determined. Providing information on how the incentive incentive value was determined, whether it was proportional to the time or effort required, and how the reward system was designed to prevent bias or undue influence can improve the transparency of the recruitment process.

14. Although the method briefly mentions the exclusion of entries with missing or incomplete information, it would be beneficial to include a more detailed description of the data quality control measures implemented throughout the study. Describing the specific criteria used to exclude incomplete or unreliable data, as well as the steps taken to verify the precision and consistency of the collected data, can help ensure the reliability and validity of the study findings.

The statistical analysis approach described appears to be comprehensive and well-structured. Some points to consider:

15. While you mentioned that you refined the model by eliminating non-significant paths, it would be beneficial to provide a more detailed explanation of the criteria you used for determining the significance of these paths. Clearly defining the thresholds or criteria used for the inclusion or exclusion of paths would enhance the transparency of the model refinement process.

16. In your description of the mediation analysis using the Bootstrap method, it would be helpful to provide more context on the specific mediators considered and the rationale behind their selection. Furthermore, explaining how the Bootstrap method was used to quantify indirect effects and the specific variables tested for mediation can add depth to the analysis process.

DISCUSSION

In the discussion section of your manuscript, you effectively summarised the findings and their implications. However, there are a few areas where the discussion could be improved:

17. While briefly touching on the potential implications of the findings, it would be beneficial to provide a more detailed discussion on how these results can be applied practically in the field. Describe how understanding the interaction between social networks, perceived social isolation, loneliness, and depressive symptoms can inform the development of targeted interventions and support programmes for people experiencing mental health challenges.

18. Highlight the novel contributions of your research in the context of the existing literature. Clearly articulate how your study contributes to the current understanding of the relationship between social networks, perceived social isolation, loneliness, and depressive symptoms, especially in the context of implications for mental health interventions. Discuss how your findings provide a nuanced understanding that can inform the development of more effective and targeted strategies to address mental health challenges related to social isolation and loneliness.

19. Based on the findings of your study, provide clear and actionable recommendations for researchers, policymakers, and mental health professionals. Discuss specific strategies and interventions that can be implemented to mitigate the adverse effects of perceived social isolation and loneliness on depressive symptoms. Include suggestions for future research directions and potential areas for further exploration in this field.

20. Although you acknowledge the limitations of the study, it would be valuable to provide a more comprehensive discussion of the implications of these limitations for the interpretation and generalisation of the results. Discuss the potential impact of cross-sectional design, self-reported data, and online data collection method on the validity and reliability of the findings. In addition, provide suggestions for future research that can address these limitations and provide more robust evidence.

REFERENCE

21. The authors included 39 references. Most of them are recent and relevant.

LANGUAGE AND GRAMMAR

22. There are issues with expressions and syntax. Ensure that the language used is clear and easily understandable to a wider audience. Try to simplify complex sentences and use straightforward language where possible.

7. PLOS authors have the option to publish the peer review history of their article (what does this mean?). If published, this will include your full peer review and any attached files.

Reviewer #1: No

Reviewer #3: **Yes: **Samir Al-Adawi

---

## [Author Response · Author response to Decision Letter 1]

30 Dec 2023

Response to academic editor and reviewers.

Journal Requirements:

1. You made some helpful revisions and good progress on this work. There are still, however, some fairly significant issues with lack of details of some methods, and statements that don't always fit with the study. Please pay particular attention to the requests from both reviewers for more details on the methodology. In addition, both reviewers mentioned statements in the manuscript that don't quite match the study content. That relates to criteria 4 here: https://journals.plos.org/plosone/s/criteria-for-publication

Response:

We reviewed the manuscript in its entirety and attempted to revise the manuscript with reference to Criterion 4 (https://journals.plos.org/plosone/s/criteria-for-publication).

Responses to Reviewer #1:

Thank you for providing critical insights that helped us to improve this paper. We have revised the manuscript and included the analytical data indicated by you. We made the following revisions based on your comments.

Abstract

Methods: How do you determine that the surveyed sample is a "representative sample"? Is it sufficiently representative? Can it represent the population of Japan?

Response:

Thank you for this comment. We have reported the analysis and results of verifying the sample's representativeness in the Result chapter, as follows.

“To assess the representativeness of our sample relative to the Japanese population, we conducted chi-square tests for gender and age, using demographic data from the Ministry of Internal Affairs and Communications as of March 2022 as a reference. Notably, our study's sample had fewer participants than expected in their 20s and 70s and more in their 40s and 50s compared to the broader Japanese population. This discrepancy suggests that our data may not entirely reflect the demographic composition of Japan. We acknowledge this as a limitation of our study and will discuss its implications in the discussion. Despite these concerns about representativeness, the collected data were utilized for further analysis because our primary objective was to explore the relationships and underlying mechanisms of the variables.” (pp12, line 209-214).

The result of our analysis indicated that our data are not representative of the Japanese population. Nevertheless, we proceeded with the subsequent analyses because this study was designed to understand the associations and mechanisms of the variables and not to determine prevalence or risk ratios in a specific population. However, we have discussed the issue of sample representativeness as a limitation of this study of online surveys.

“Lastly, the online nature of data collection might lead to sampling bias. The data might not fully represent the Japanese population, notably missing those without internet access or experiencing severe social isolation or loneliness. To enhance these findings, further research should consider conducting in-depth interviews with people facing social isolation or intense loneliness (e.g., individuals who have withdrawn from society) and demographic studies in specific communities” (pp19, line 334-339).

Conclusions: What are the intervention measures? Please write the key conclusion based on your question: "How can we reduce the effects of social isolation and loneliness on compressive symptoms".

Response:

Thank you for these comments. Based on your comment, I have included the following critical conclusions in the revised manuscript.

“Interventions to effectively decrease social isolation and loneliness and thereby mitigate their impact on depressive symptoms should include cognitive behavioral therapy techniques. These techniques aim to alter negative perceptions of social isolation and loneliness and bolster interpersonal skills. We anticipate future longitudinal studies to further refine and strengthen the model proposed in this study by clarifying causal relations among these variables.” (pp3, line 26-30).

Methods

1.Participants and Procedures: You still haven't solved the core problem. 1) In the research design stage, the size of the sample size should be calculated. Secondly, how is the study population determined? Why choose this research population? 3) Didn't even collect demographic information?

Response:

Thank you for your comments. In the revised manuscript, we have included information on determining the sample size and study population. We also created Table 1 to show the participants’ demographic information.

“We decided to collaborate with Cross Marketing Inc. because it is the most prominent research company in Japan. The sample size for our study was determined using G*Power 3.1.9.7 [40, 41]. Since the purpose of this study was to examine the hypothetical model in an exploratory manner, the sample size required was calculated to be 779 cases, assuming a two-tailed test for the mother correlation coefficient, an assumed correlation coefficient of 0.10, a significance level of 5%, and a power of 80%. We also planned to conduct structural equation modeling (SEM). However, there is no consensus on pre-calculating the power and sample size for SEM [42]. It is recognized that a small sample size may compromise the accuracy and reproducibility of results of SEM [43]. Therefore, we aimed for the largest feasible within our funding constraints, exceeding 779 participants, to ensure a robust analysis and to capture trends representative of the Japanese population.” (pp8, line 118-131).

Please see the Table 1. Respondents’ characteristics in the revised manuscript. (pp13)

2.Measures, Lines 112-117: You have not made any modifications yet. You should add data that can demonstrate the reliability and validity of the scale, and this data should have references.

Response:

Thank you for your suggestions. I have added the data showing the reliability and validity of the LSNS-6. We assessed the reliability using Cronbach's alpha and validity using confirmatory factor analysis as a two-factor structure, similar to previous studies (Kurimoto et al., 2011). We hope the following information is helpful.

“We first conducted a chi-square test on sociodemographic variables to ensure the sample's representativeness. Next, we conducted a confirmatory factor analysis (CFA) to verify structural validity of the LSNS-6. Based on previous research [47], we analyzed a model where the LSNS-6 Family and LSNS-6 Friend factors were each represented by three items. The model's goodness of fit was assessed using several metrics: the chi-square (χ2) test statistic, the Root Mean Squared Error of Approximation (RMSEA), the Comparative Fit Index (CFI), and the Tucker-Lewis Index (TLI). A non-significant χ2 test statistic implies an acceptable fit, but it is essential to note its sensitivity in large sample sizes [48]. CFI and TLI values above 0.95 indicate a good model fit [49], whereas RMSEA values below 0.05 suggest an optimal fit, values between 0.05 and 0.08 a moderate fit, and values between 0.08 and 0.10 a marginal fit [50]. The reliability of each scale was determined using Cronbach's alpha, a widely recognized measure of reliability [51]. An alpha value of 0.70 or higher is considered acceptable [52].” (pp10, line 171-182)

“We conducted a confirmatory factor analysis to test the structural validity of the LSNS-6. The model demonstrated a good fit, evidenced by χ2 (8) = 81.03, p < .001, with RMSEA = .05, CFI = .99, and TLI = .99. This confirmed the validity of the two-factor structure for our study's data. We calculated the Cronbach's alpha coefficient to be 0.88 for LSNS-6 Family and 0.89 for LSNS-6 Friend, indicating appropriate reliability for both scales.” (pp13, line 218-222).

Responses to Reviewer #3:

Thank you for providing critical insights that helped us to improve this paper. We have revised the manuscript and included the analytical data indicated by you. We made the following revisions based on your comments.

I appreciate the opportunity to review the manuscript titled "How can we reduce the effects of social isolation and loneliness on depressive symptoms: Findings from an online survey of a Japanese population." This study investigates the correlation between social isolation, loneliness, and perceived social isolation in connection with depressive symptoms in the adult Japanese population. The authors conducted an online survey among 3315 representative Japanese adults to investigate these connections. The survey incorporated the Lubben Social Network Scale to gauge the size of the participants' social networks, the UCLA Loneliness Scale consisting of 3 items to evaluate loneliness, and the Patient Health Questionnaire-9 to measure depressive symptoms. In particular, the extensive sample size of 3315 participants enhances the robustness of the findings. The authors employed structural equation modelling to examine their hypothetical model, which considered the impact of social networks, perception of social isolation, and loneliness on depressive symptoms. The results revealed that larger social networks exhibited a weak correlation with lower levels of loneliness and depressive symptoms. Specifically, the study indicated that perceived social isolation was associated with increased levels of both loneliness and depressive symptoms.

In general, this study has considerable merit. To further enrich the scientific value of the manuscript, I have outlined some key points for the authors' consideration based on the feedback from the two reviewers. My aim is to ensure a comprehensive understanding of the comments of the reviewers, to evaluate the adequacy of the authors' responses, and to provide additional suggestions to enhance the manuscript.

Response:

Thank you very much for your evaluation of this paper. We appreciate the opportunity to revise the manuscript in detail. We have modified the entire manuscript as you suggested. Based on your comments, we have made the following revisions.

TITLE

1. While the current title of the study provides some information on its focus, I would recommend a more descriptive one such as "Examining the Impact of Social Networks and Perceived Social Isolation on Depressive Symptoms among Adults in Japan". At its current stage, it would be premature to conclude that the study aims to mitigate the negative impact of social isolation and loneliness on depressive symptoms.

Response:

Thank you for your suggestion. As you suggested, we changed the title to be more descriptive.

“How Social Networks, Social Isolation and Loneliness Effect on Depressive Symptoms among Japanese Adults?” (pp1, line 4-6).

ABSTRACT

The abstract provides a comprehensive overview of the study. However, a few improvements could enhance its clarity and impact.

2. Please consider changing (“We assessed the size of the social network size using the Lubben Social Network Scale”) to (“We used the Lubben Social Network Scale to determine the size of the social network”).

Response:

Thank you for your comments. We have revised the abstract according to your suggestion.

“We used the six-item Lubben Social Network Scale to assess the size of their social networks.” (pp2, line 9-10)

3. The statistical significance of the goodness of fit of the model should be emphasized. Providing a brief interpretation of the beta coefficients and their implications for the interaction between social isolation, loneliness, and depressive symptoms would make the results more accessible to a wider audience.

Response:

Thank you for your valuable advice. In the revised manuscript, I described that the model had a good fit, as shown below.

“The final model demonstrated a satisfactory fit with the data (χ2 (1) = 3.73; not significant; RMSEA = 0.03; CFI = 1.00; TLI = 1.00).” (pp2, line 16-17).

We also determined that it is appropriate to include the interpretation of the beta, or path coefficient, in the statistical analysis section described in the text below. We also stated that the chi-square value is an indicator of model fit sensitive to the sample size.

“A non-significant χ2 test statistic implies an acceptable fit, but it is essential to note its sensitivity in large sample sizes [48]. CFI and TLI values above 0.95 indicate a good model fit [49], whereas RMSEA values below 0.05 suggest an optimal fit, values between 0.05 and 0.08 a moderate fit, and values between 0.08 and 0.10 a marginal fit [50].” (pp10, line 176-180)

4. The conclusion could be strengthened by reiterating the practical implications of the study findings and how they can contribute to the development of effective intervention strategies. Furthermore, providing some suggestions for future research directions can add depth to the conclusion.

Response:

Thank you for your comments. We have revised our conclusions as follows.

“Interventions to effectively decrease social isolation and loneliness and thereby mitigate their impact on depressive symptoms should include cognitive behavioral therapy techniques. These techniques aim to alter negative perceptions of social isolation and loneliness and bolster interpersonal skills. We anticipate future longitudinal studies to further refine and strengthen the model proposed in this study by clarifying causal relations among these variables.” (pp3, line 26-30)

In the discussion, we included details on the practical implications of this study and its contribution to developing effective strategies.

“However, studies on the effectiveness of such preventive measures are limited, underscoring the need to develop intervention programs focusing on primary prevention against social isolation and loneliness. Our study suggests developing early psychoeducational programs for school education and community-based cognitive-behavioral therapeutic interventions to prevent social isolation and loneliness. Such initiatives could play a vital role in preventing depressive symptoms arising from social isolation and loneliness.” (pp19, line 322-327)

INTRODUCTION

5. Although the study's hypothetical model is briefly mentioned, providing a more detailed integration of the model within the context of the existing literature could strengthen the background and set the stage for the study's aims and objectives.

Response:

Thank you for your very useful suggestions. I have added a more detailed explanation of the hypothetical model and included the reasons for using perceived social isolation and loneliness as mediating factors.

“From existing literature, we hypothesize that having more of social networks is negatively related to perception of social isolation, loneliness, and depressive symptoms [24, 27, 28]. We expected that perceptions of social isolation, which arise from individuals evaluating their social networks, positively related to loneliness and depressive symptoms [24, 25, 34]. Consistent with its established link to depression [29], we expected loneliness to relate positively to depressive symptoms. Further, we suggest that perceptions of social isolation and loneliness might mediate the relationship between social networks and depressive symptoms, as these perceptions often have a significant impact on mental health than objective social isolation measure [25, 30, 31].” (pp6, line 97-104)

6. I suggest narrating the terms in the text and providing relevant literature instead of intruding on a table to define variables, as it breaks the conventional approach.

Response:

Thank you for your suggestion. We have removed Table 1 and modified it to include subheadings to explain the terms in the text with related references.

7. The relevance of this entry is not clear (Fig 1. Hypothetical model of this study. The squares indicate a psychological variable and the 1 arrows indicate associations among variables.)

Response:

Thank you for pointing this out. We have revised Figure 1 to reflect the positive and negative predictions of the path coefficients between each variable.

 Fig 1. Hypothetical model of this study. In this diagram, squares represent different psychological variables. Arrows between these squares depict the relationships among these variables. Dashed arrows illustrate negative, and solid arrows indicate positive paths.

8. The authors did mention some of the expressions of loneliness, including hikikomori. There is an additional concept that correlates with loneliness that needs to be mentioned: social avoidant personality disorder, Taijin kyofusho, Modern Type Depression, Unemployment, and Not in Employment Education or Training (NEET). The economic and demographic factors that contribute to increased loneliness in Japan need to be further elaborated.

Response:

Thank you for your thought-provoking comment. I have included additional information on the concepts and economic and demographic factors contributing to the increase in social isolation and loneliness in Japan.

“Factors such as social anxiety, avoidant personality disorder, and modern depression increasingly linked to heightened social isolation and loneliness in Japan [3-5], where these issues are more pronounced than in other cultures [6, 7]. Specifically, social anxiety disorder and avoidant personality disorder are believed to lead to societal withdrawal, especially among young people [8]. Additionally, the rise of individuals Not in Employment Education or Training (NEET) and unemployment -related economic challenges are associated with increasing social isolation and loneliness in Japan [9, 10].” (pp4, line 38-44)

METHOD

9. In the abstract, we were told 'Lubben Social Network Scale… with a 1-item question' and in the method, we were told (“The LSNS-6 comprises six items…”.). Please reconcile.

Response:

Thank you for pointing this out. I made the following revisions to maintain consistency.

“We used the six-item Lubben Social Network Scale to assess the size of their social networks. We employed a single question to gauge perceived social isolation. Loneliness was assessed using the three-item UCLA Loneliness Scale, and depressive symptoms were assessed using the Patient Health Questionnaire-9.” (pp2, line 9-12)

10. The authors have described the outcome measures under the subheading "Measures". However, it may be necessary to mention the psychometric properties of the Japanese version of these measures. On the other hand, in line 131, pg. 6, it was stated that “Initially, we computed descriptive statistics for LSNS-6 Family, LSNS-6 Friend, Perception of Social Isolation, R-UCLA, and PHQ-9. We evaluated the reliability of each scale by determining Cronbach's alpha, a widely recognized measure of reliability [36]. An alpha value of 0.70 or higher is deemed acceptable [37]”. Exercises to explore reliability should be detailed in the subheading ‘Measures’ and removed from the subheading ‘statistical analyses. Some of the literature cited appears to allude that the Japanese reliability versions of these instruments. Indeed, the authors did cite the relevant study but did not narrate the issue of the Japanese version of the LSNS-6 (Kurimoto A, Awata S, Ohkubo T, Tsubota-Utsugi M, Asayama K, Takahashi K, Suenaga K, Satoh H, Imai Y. [Reliability and validity of the Japanese version of the abbreviated Lubben Social Network Scale]. Nihon Ronen Igakkai Zasshi. 2011;48(2):149-57. Japanese. doi: 10.3143/geriatrics.48.149.

Response:

Thank you for your comments, we have added the confirmatory factor analysis data to the Results section to demonstrate the reliability and validity of the LSNS-6. We have also included Cronbach's alpha coefficients for each scale in the Results section, as we believe it is appropriate to include Cronbach's alpha coefficients for assessing reliability along with the results of the confirmatory factor analysis.

“We first conducted a chi-square test on sociodemographic variables to ensure the sample's representativeness. Next, we conducted a confirmatory factor analysis (CFA) to verify structural validity of the LSNS-6. Based on previous research [47], we analyzed a model where the LSNS-6 Family and LSNS-6 Friend factors were each represented by three items. The model's goodness of fit was assessed using several metrics: the chi-square (χ2) test statistic, the Root Mean Squared Error of Approximation (RMSEA), the Comparative Fit Index (CFI), and the Tucker-Lewis Index (TLI). A non-significant χ2 test statistic implies an acceptable fit, but it is essential to note its sensitivity in large sample sizes [48]. CFI and TLI values above 0.95 indicate a good model fit [49], whereas RMSEA values below 0.05 suggest an optimal fit, values between 0.05 and 0.08 a moderate fit, and values between 0.08 and 0.10 a marginal fit [50]. The reliability of each scale was determined using Cronbach's alpha, a widely recognized measure of reliability [51]. An alpha value of 0.70 or higher is considered acceptable [52].” (pp10, line 171-182)

“We conducted a confirmatory factor analysis to test the structural validity of the LSNS-6. The model demonstrated a good fit, evidenced by χ2 (8) = 81.03, p < .001, with RMSEA = .05, CFI = .99, and TLI = .99. This confirmed the validity of the two-factor structure for our study's data. We calculated the Cronbach's alpha coefficient to be 0.88 for LSNS-6 Family and 0.89 for LSNS-6 Friend, indicating appropriate reliability for both scales. Also, the Cronbach's alpha coefficients for R-UCLA and PHQ-9 were 0.85 and 0.92, respectively, both of which were sufficiently reliable.” (pp13, line 218-223)

11. Although the method states that efforts were made to ensure diversity in the participant pool, it would be beneficial to provide more explicit details on the specific measures taken to ensure representation across various sociodemographic factors, such as age, location (rural and urban), socioeconomic status, and cultural background. Including a breakdown of these demographics in the participant pool can add credibility to the study's results.

Response:

Thank you for your critical comments. We have presented the demographic data in Table 1.

Please see the Table 1. Respondents’ characteristics in revised manuscript. (pp13)

12. Although the method briefly mentions that participants were informed about the study objectives and ethical considerations, it would be advantageous to provide more detailed information on the informed consent process. Specifically, explaining how participants were informed about their rights, the purpose of the study, the possible risks and benefits, and how their data would be used and protected can help ensure transparency and build trust with participants.

Response:

Thank you for your suggestion. We have added details of the informed consent procedure as follows.

“Before commencing the survey, they were informed that their responses would remain anonymous and that there would be no negative consequences for choosing not to respond or withdrawing from the survey at any point. It was explained that all survey responses would be processed statistically, ensuring no individual identification. Furthermore, strict protocols were followed in handling all responses, from the processing stage to data storage and eventual disposal. Participation was entirely voluntary and based on the informed consent of the participants.” (pp8, line 133-138)

13. Although the method mentions that participants were given reward points as a token of appreciation, it might be useful to elaborate on the nature of these incentives and how they were determined. Providing information on how the incentive value was determined, whether it was proportional to the time or effort required, and how the reward system was designed to prevent bias or undue influence can improve the transparency of the recruitment process.

Response:

Thank you for your comment. I have described the reward points given to participants.

“They received reward points from the survey agency as a gesture of gratitude for their time and input. These points, determined by the number of questions answered, are a unique feature of Cross Marketing and could be exchanged by participants for goods or cash.” (pp8-9, line 139-141)

14. Although the method briefly mentions the exclusion of entries with missing or incomplete information, it would be beneficial to include a more detailed description of the data quality control measures implemented throughout the study. Describing the specific criteria used to exclude incomplete or unreliable data, as well as the steps taken to verify the precision and consistency of the collected data, can help ensure the reliability and validity of the study findings.

Response:

Thank you for the valuable suggestion. I have described the criteria for excluding incomplete data as follows.

“The criteria for exclusion included responses that were evidently dishonest (for example, identical numerical answers for all survey items) and missing responses to the scales measuring social networks, perceptions of social isolation, loneliness, and depressive symptoms.” (pp9, line 143-145)

The statistical analysis approach described appears to be comprehensive and well-structured. Some points to consider:

15. While you mentioned that you refined the model by eliminating non-significant paths, it would be beneficial to provide a more detailed explanation of the criteria you used for determining the significance of these paths. Clearly defining the thresholds or criteria used for the inclusion or exclusion of paths would enhance the transparency of the model refinement process.

Response:

Thank you for the useful comment. We described the exclusion criteria for non-significant paths as shown below. We also described the excluded paths from the hypothetical model through structural equation modeling.

“The threshold for statistical significance of path coefficients was set at 5%. Our analysis revealed that the path linking social isolation from friends to depressive symptoms was not statistically significant, leading to its exclusion from the model.” (pp14, line 242-245)

16. In your description of the mediation analysis using the Bootstrap method, it would be helpful to provide more context on the specific mediators considered and the rationale behind their selection. Furthermore, explaining how the Bootstrap method was used to quantify indirect effects and the specific variables tested for mediation can add depth to the analysis process.

Response:

Thank you for this comment. I added a detailed explanation of the bootstrap method to the statistical analysis section.

“We also performed a mediation analysis using the Bootstrap method to assess the indirect effects within the SEM framework. This method takes a sample of researchers of size N and creates a new sample from which to extract the replacement N values of the independent, mediating, and dependent variables. For example, this option can be repeated 5000 times to compute 5000 indirect effect estimates [54]. The Bootstrap method is known for providing accurate confidence intervals for indirect effects, as these are based on the empirical distribution of the estimates [55, 56].” (pp11, line 198-202)

In addition, I provided an explanation of the rationale for considering social isolation and loneliness as mediating factors in the Introduction.

“Further, we suggest that perceptions of social isolation and loneliness might mediate the relationship between social networks and depressive symptoms, as these perceptions often have a significant impact on mental health than objective social isolation measure [25, 30, 31].” (pp6, line 101-104)

DISCUSSION

In the discussion section of your manuscript, you effectively summarised the findings and their implications. However, there are a few areas where the discussion could be improved:

17. While briefly touching on the potential implications of the findings, it would be beneficial to provide a more detailed discussion on how these results can be applied practically in the field. Describe how understanding the interaction between social networks, perceived social isolation, loneliness, and depressive symptoms can inform the development of targeted interventions and support programs for people experiencing mental health challenges.

Response:

Thank you for your comment. We have reiterated the mechanisms identified in this study and included a discussion of the potential application of these results.

“Our findings revealed that the direct effect of social network size on loneliness and depressive symptoms was marginal. However, comprehensively analyzing social networks, perceptions of social isolation, and loneliness and examining their effects on depressive symptoms identified a mechanism in which the relationship between social network size and depressive symptoms was mediated by the perceptions of social isolation and loneliness that increase feelings of loneliness and depressive symptoms. This result extends the findings of previous studies that examined these variables in isolation [24-26, 29-34].” (pp18, line 288-294)

“While most studies on social isolation and loneliness have focused on older adults [37], the findings of this study are significant for showing that social isolation and loneliness can cause mental health decline, regardless of age or gender, underscoring the universal applicability of our findings.” (pp18, line 299-302)

18. Highlight the novel contributions of your research in the context of the existing literature. Clearly articulate how your study contributes to the current understanding of the relationship between social networks, perceived social isolation, loneliness, and depressive symptoms, especially in the context of implications for mental health interventions. Discuss how your findings provide a nuanced understanding that can inform the development of more effective and targeted strategies to address mental health challenges related to social isolation and loneliness.

Response:

Thank you for your valuable comment. We have revised our discussion to describe the novelty of this study clearly. We have also described the implications of our findings for developing strategies and programs for social isolation and loneliness.

However, comprehensively analyzing social networks, perceptions of social isolation, and loneliness and examining their effects on depressive symptoms identified a mechanism in which the relationship between social network size and depressive symptoms was mediated by the perceptions of social isolation and loneliness that increase feelings of loneliness and depressive symptoms. This result extends the findings of previous studies that examined these variables in isolation [24-26, 29-34].” (pp18, line 289-294)

“While most studies and intervention methods on social isolation and loneliness have focused on older adults [37, 57, 58], the findings of this study are significant for showing that social isolation and loneliness can cause mental health decline, regardless of age or gender, underscoring the universal applicability of our findings.” (pp18, line 299-302)

19. Based on the findings of your study, provide clear and actionable recommendations for researchers, policymakers, and mental health professionals. Discuss specific strategies and interventions that can be implemented to mitigate the adverse effects of perceived social isolation and loneliness on depressive symptoms. Include suggestions for future research directions and potential areas for further exploration in this field.

Response:

Thank you for your comment. Based on the research findings, we have described specific interventions and strategies to reduce or prevent the impact of social isolation and loneliness on depressive symptoms.

“Our findings underscore the importance of addressing the transition from perceptions of social isolation and loneliness to depression. Interventions aimed at altering perceptions of social isolation can help reduce depressive symptoms and alleviate loneliness if they target cognitive and emotional aspects across various age groups. Psychoeducational approaches and cognitive-behavioral therapy for changing cognitions and emotions alleviate loneliness effectively [36]. These interventions can be pivotal in transforming perceptions and reducing loneliness and depressive symptoms. Additionally, reminiscence-based psychological interventions have shown effectiveness [57]. We advocate for strategies focusing on the cognitive and emotional facets of social isolation and loneliness rather than merely expanding social networks [39] because social isolation does not necessarily lead to loneliness [23].

Previous research underscores the importance of primary prevention in addressing social isolation and loneliness [60-62]. However, studies on the effectiveness of such preventive measures are limited, underscoring the need to develop intervention programs focusing on primary prevention against social isolation and loneliness. Our study suggests developing early psychoeducational programs for school education and community-based cognitive-behavioral therapeutic interventions to prevent social isolation and loneliness. Such initiatives could play a vital role in preventing depressive symptoms arising from social isolation and loneliness.” (pp18-19, line 311-327)

20. Although you acknowledge the limitations of the study, it would be valuable to provide a more comprehensive discussion of the implications of these limitations for the interpretation and generalization of the results. Discuss the potential impact of cross-sectional design, self-reported data, and online data collection method on the validity and reliability of the findings. In addition, provide suggestions for future research that can address these limitations and provide more robust evidence.

Response:

Thank you for pointing out the limitations of this study. We have discussed the limitations of interpreting and generalizing the results. We have also added suggestions for future studies to provide more robust evidence.

“Lastly, the online nature of data collection might lead to sampling bias. The data might not fully represent the Japanese population, notably missing those without internet access or experiencing severe social isolation or loneliness. To enhance these findings, further research should consider conducting in-depth interviews with people facing social isolation or intense loneliness (e.g., individuals who have withdrawn from society) and demographic studies in specific communities.” (pp19, line 334-339)

REFERENCE

21. The authors included 39 references. Most of them are recent and relevant.

Thank you for your observation. We have included more references as we revised the manuscript.

LANGUAGE AND GRAMMAR

22. There are issues with expressions and syntax. Ensure that the language used is clear and easily understandable to a wider audience. Try to simplify complex sentences and use straightforward language where possible.

Thank you for taking the time to review our manuscript. We have revised the English expressions and syntax in the text to make it as straightforward as possible so that many readers can easily understand it.

We thank you again for allowing us to improve our manuscript with your valuable comments and queries. We have worked hard to incorporate your feedback and hope these revisions will persuade you to accept our manuscript for publication in PLOS ONE.

---

## [Decision Letter · Decision Letter 2]

25 Jan 2024

PONE-D-23-09807R2How social networks, social isolation and loneliness effect on depressive symptoms among Japanese adults?PLOS ONE

Dear Dr. Tachikawa,

Thank you for submitting your manuscript to PLOS ONE. After careful consideration, we feel that it has merit but does not fully meet PLOS ONE’s publication criteria as it currently stands. Therefore, we invite you to submit a revised version of the manuscript that addresses the points raised during the review process.

We look forward to receiving your revised manuscript.

Kind regards,

Keith M. Harris, PhD

Academic Editor

PLOS ONE

Journal Requirements:

Additional Editor Comments:

Thank you for addressing feedback well on this revised manuscript. There is some need for language revision to improve flow and readability. Some editing by a native speaker appears all that is needed. That will only require an editor and not a full review.

Reviewers' comments:

Reviewer's Responses to Questions

**Comments to the Author**

1. If the authors have adequately addressed your comments raised in a previous round of review and you feel that this manuscript is now acceptable for publication, you may indicate that here to bypass the “Comments to the Author” section, enter your conflict of interest statement in the “Confidential to Editor” section, and submit your "Accept" recommendation.

Reviewer #1: (No Response)

Reviewer #3: All comments have been addressed

2. Is the manuscript technically sound, and do the data support the conclusions?

Reviewer #1: (No Response)

Reviewer #3: Yes

3. Has the statistical analysis been performed appropriately and rigorously? 

Reviewer #1: Yes

Reviewer #3: Yes

4. Have the authors made all data underlying the findings in their manuscript fully available?

Reviewer #1: Yes

Reviewer #3: Yes

5. Is the manuscript presented in an intelligible fashion and written in standard English?

Reviewer #1: No

Reviewer #3: Yes

6. Review Comments to the Author

Reviewer #1: ID: PONE-D-23-09807R2

Title: How social networks, social isolation and loneliness effect on depressive symptoms among Japanese adults?

Thank you for providing a chance to review this manuscript.

Recommendation: Minor revision.

Detailed information:

Abstract

Conclusions: Please use shorter language to explain, this is an abstract and not a summary.

There are no other objections, however, this paper needs to be thoroughly edited by native English-speakers.

Thank you and my best,

Your reviewer

Reviewer #3: Thank you for reconsidering to re-review the manuscript titled "How Social Networks, Social Isolation, and Loneliness Affect Depressive Symptoms Among Japanese Adults." I appreciate that the authors have incorporated all of my suggestions, and overall the manuscript is commendable. It seems ready for acceptance with only a few minor revisions. Great job!

TITLE

The title is generally clear and conveys the main topics of the study. However, there are a couple of grammatical issues that could be addressed for clarity. Use of "effect" vs. "affect": The correct term to use in this context is "affect" rather than "effect." "Affect" is a verb that means to produce a change or influence, while "effect" is a noun referring to the change or result in itself. Therefore, the title should be: Revised Title: "How Social Networks, Social Isolation, and Loneliness Affect Depressive Symptoms among Japanese Adults."

ABSTRACT

For consistency, you may want to ensure that you use similar terms throughout the title. For instance, you used "loneliness," but in the abstract, the term "perceived social isolation" is mentioned. Depending on the specific focus of your study, you might want to use consistent terminology.

INTRODUCTION

Briefly elaborate on the "8050 problem" to provide readers with a clearer understanding of this specific societal challenge in Japan.

Otherwise, it should be ready to be accepted and may have a more thorough check of grammar and syntax before publication.

7. PLOS authors have the option to publish the peer review history of their article (what does this mean?). If published, this will include your full peer review and any attached files.

Reviewer #1: No

Reviewer #3: **Yes: **Samir Al-Adawi

---

## [Author Response · Author response to Decision Letter 2]

25 Feb 2024

Additional Editor Comments:

Thank you for addressing feedback well on this revised manuscript. There is some need for language revision to improve flow and readability. Some editing by a native speaker appears all that is needed. That will only require an editor and not a full review.

Response:

Thank you for your appreciation of our response. The manuscript was proofread by a native speaker to improve flow and readability. We have also attached a proofreading certificate for your review.

Responses to Reviewer #1:

Thank you for reviewing our revised manuscript. We have made the following revisions to the manuscript with regard to the areas you have indicated.

Abstract

Conclusions: Please use shorter language to explain, this is an abstract and not a summary.

Response:

Thank you for pointing this out. We have revised the description to be shorter.

“Results indicate that interventions of psychological approaches, such as cognitive–behavioral therapy, are effective in reducing the perception of social isolation and loneliness, which may lead to the prevention of depressive symptoms. Future longitudinal studies are expected to refine and strengthen the proposed model.” (pp2, line 21-24).

There are no other objections, however, this paper needs to be thoroughly edited by native English-speakers.

Response:

Thank you for your confirmation. The manuscript has been entirely proofread by a native speaker and the problem areas have been corrected. We have also attached a proofreading certificate for your review.

Responses to Reviewer #3:

Thank you for reconsidering to re-review the manuscript titled "How Social Networks, Social Isolation, and Loneliness Affect Depressive Symptoms Among Japanese Adults." I appreciate that the authors have incorporated all of my suggestions, and overall the manuscript is commendable. It seems ready for acceptance with only a few minor revisions. Great job!

Response:

Thank you very much for evaluating our work. We deeply appreciate your suggestions to further improve our manuscript. We have revised the manuscript as follows with regard to the points you have highlighted.

TITLE

The title is generally clear and conveys the main topics of the study. However, there are a couple of grammatical issues that could be addressed for clarity. Use of "effect" vs. "affect": The correct term to use in this context is "affect" rather than "effect." "Affect" is a verb that means to produce a change or influence, while "effect" is a noun referring to the change or result in itself. Therefore, the title should be: Revised Title: "How Social Networks, Social Isolation, and Loneliness Affect Depressive Symptoms among Japanese Adults."

Response:

Thank you for highlighting this grammatical problem. We have revised the title as per your suggestion.

“How Do Social Networks, Perception of Social Isolation, and Loneliness Affect Depressive Symptoms among Japanese Adults?”

ABSTRACT

For consistency, you may want to ensure that you use similar terms throughout the title. For instance, you used "loneliness," but in the abstract, the term "perceived social isolation" is mentioned. Depending on the specific focus of your study, you might want to use consistent terminology.

Response:

Thank you for your valuable suggestions. In our research, we treat “perceived social isolation” and “loneliness” differently. Therefore, we have changed the term “social isolation” in the title to “Perception of social isolation”.

“How Do Social Networks, Perception of Social Isolation, and Loneliness Affect Depressive Symptoms among Japanese Adults?”

In addition, the terms “perceptions of social isolation” and “perceived social isolation” in the abstract were revised to “perception of social isolation” to make them consistent.

“This study aims to elucidate the complex relationship among social isolation, loneliness, and perception of social isolation and its influence on depressive symptoms by evaluating a hypothetical model.”（pp2, line3-5）

“We employed a single question to gauge perception of social isolation.”（pp2, line8-9）

INTRODUCTION

Briefly elaborate on the "8050 problem" to provide readers with a clearer understanding of this specific societal challenge in Japan.

Response:

Thank you for your suggestion. We have added an explanation of the 8050 issue.

“A particularly acute issue, which is known as the 8050 problem, involves octogenarian parents caring for their socially reclusive children aged in their 50s [1]. In this problem, socially withdrawn children become middle-aged, and the parents who have cared for them typically become elderly, such that caring for their children becomes increasingly difficult. This issue is not unique to Japan; instead, it is a grave one internationally [2] and measures are required to address this issue.” （pp3, line29-34）

Otherwise, it should be ready to be accepted and may have a more thorough check of grammar and syntax before publication.

Response:

Thank you for your careful review of our manuscript. It was thoroughly proofread by a native English speaker for grammar and syntax. We hope you will also review the proofreading certificate.

---

## [Editor Report · Decision Letter 3]

27 Feb 2024

How Do Social Networks, Perception of Social Isolation, and Loneliness Affect Depressive Symptoms among Japanese Adults?

PONE-D-23-09807R3

Dear Dr. Tachikawa,

We’re pleased to inform you that your manuscript has been judged scientifically suitable for publication and will be formally accepted for publication once it meets all outstanding technical requirements.

Kind regards,

Keith M. Harris, PhD

Academic Editor

PLOS ONE

Additional Editor Comments (optional):

Thank you for taking the time to rework the writing of the manuscript. I see you made quite a few minor revisions, that to me led to a very easy to read and more concise work. Best wishes in your further studies!
---

## [Editor Report · Acceptance letter]

28 Mar 2024

PONE-D-23-09807R3 

PLOS ONE

Dear Dr. Tachikawa, 

I'm pleased to inform you that your manuscript has been deemed suitable for publication in PLOS ONE. Congratulations! Your manuscript is now being handed over to our production team.

Kind regards, 

on behalf of

Dr. Keith M. Harris 

Academic Editor

PLOS ONE